# Open-Vocabulary Audio-Visual Semantic Segmentation

## ABSTRACT

Audio-visual semantic segmentation (AVSS) aims to segment and classify sounding objects in videos with acoustic cues. However, most approaches operate on the close-set assumption and only identify pre-defined categories from training data, lacking the generalization ability to detect novel categories in practical applications. In this paper, we introduce a new task: **open-vocabulary audio-visual semantic segmentation**, extending AVSS task to open-world scenarios beyond the annotated label space. This is a more challenging task that requires recognizing all categories, even those that have never been seen nor heard during training. Moreover, we propose the first open-vocabulary AVSS framework, **OV-AVSS**, which mainly consists of two parts: 1) a universal sound source localization module to perform audio-visual fusion and locate all potential sounding objects and 2) an open-vocabulary classification module to predict categories with the help of the prior knowledge from large-scale pre-trained vision-language models. To properly evaluate the open-vocabulary AVSS, we split zero-shot training and testing subsets based on the AVSBench-semantic benchmark, namely **AVSBench-OV**. Extensive experiments demonstrate the strong segmentation and zero-shot generalization ability of our model on all categories. On the AVSBench-OV dataset, OV-AVSS achieves 55.43% mIoU on base categories and 29.14% mIoU on novel categories, exceeding the state-of-the-art zero-shot method by 41.88%/20.61% and open-vocabulary method by 10.2%/11.6%.

## CCS CONCEPTS

• **Computing methodologies** → **Video segmentation**.

## KEYWORDS

Open-Vocabulary Learning, Audio-Visual Semantic Segmentation, Vision-Language Model, Transformer, Multi-Modal Fusion

## 1 INTRODUCTION

Generally speaking, objects can be characterized jointly by their appearance and the sounds they make. By incorporating visual and acoustic information in a collaborative manner, it is beneficial for a better perception and understanding of the concepts of objects. Recently, a novel audio-visual learning task, i.e., audio-visual segmentation (AVS), has been proposed in [37]. The goal of AVS is to output pixel-level maps of sound-emitting objects within video frames. This requires aligning two different modalities and serving audio as prompt to localize and segment visual

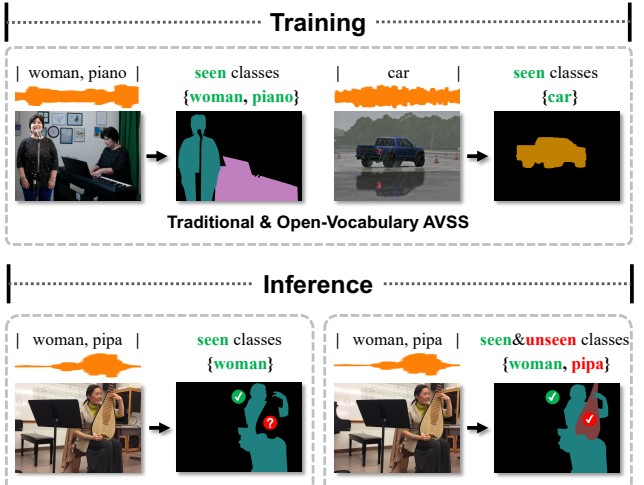

**Figure 1: An illustration of open-vocabulary audio-visual semantic segmentation. (a) Traditional AVSS models trained on closed-set classes (woman, piano, and car) fail to segment novel class (pipa). (b) Our open-vocabulary model correctly localizes sounding objects and recognizes arbitrary categories, e.g., pipa, without using any annotations.**

objects. Different from AVS being tasked with binary foreground segmentation, audio-visual semantic segmentation (AVSS) is to classify sound sources and generate semantic maps associating one category with each pixel. To accomplish the above tasks, many works adopt convolutional- or Transformer-based encoder-decoder architectures to establish audio-visual relationships and segment sounding objects. Despite promising results, these approaches only recognize pre-defined categories in the training set, resulting in poor generalization capacity to novel concepts. In real-world applications, this closed-set paradigm lacks practical value, since it often encounters objects from novel categories during training.

*If the model has never seen an object and heard its sound, could it still accurately localize sound source in audible videos?* Wang *et al.* [28] present a prompt-based model, GAVS, which attempts to pinpoint sounding objects real-world videos, including novel categories unseen during training. Specifically, they insert Adapter [14] into the large-scale pre-trained segment-anything model (SAM) [15] to construct audio-visual correlations and learn downstream tasks. With the help of the adapter-based fine-tunning technique, GAVS adapts the visual foundational model, i.e., SAM, to the AVS task, and generalize the prior knowledge to unseen objects and different datasets. However, GAVS is designed for binary segmentation, that is, it can only localize visual position of sounding objects without predicting their categories. In addition, GAVS takes only an image along with one-second audio as input and fuse multi-modal in spatial domains, leading to the inability to capture frame-by-frame

relationships in temporal domains. Yu *et al.* [33] explore the feasibility of promoting visual foundational models for the AVS task in a zero-shot setting. They use CLAP [30], a large-scale audio-language pre-trained model, to classify the input audio from open-set categories. Then, the predicted class names are fed into Grounding DINO [20] to obtain box predictions, and these boxes prompt SAM [15] to generate the corresponding masks. Nevertheless, only using audio information to predict the category of sounding objects is weak and inadequate in complex scenarios. For example, humans can imitate cat meows, and cars and airplanes can produce similar engine sounds. Moreover, due to the lack of visual appearances and motion cues, this model tends to segment all objects belonging to the predicted class instead of those sounding objects.

In this paper, we propose a novel multi-modal task, i.e., open-vocabulary audio-visual semantic segmentation, which aims at segmenting and classifying sound-emitting objects in videos from open-set categories, as illustrated in Fig. 1. Unlike the above zero-shot AVS methods that generate binary masks, we focus on the semantic segmentation and require the network to assign semantic labels to each pixel in given video frames. Open-vocabulary audio-visual semantic segmentation is a more challenging task, facing the following problems: ❶ Audio is a 1D signal that exhibits high information density, meaning that multiple objects could make sound simultaneously and their audio could entangle at any timestamp. Thus, it is not easy to build correct spatial alignment from mixed-source audio to visual content. ❷ Video is continuous and only using a single image-audio pair is not optimal. Temporal information and motion cues play an indispensable role in sound source localization and classification. So how to effectively integrate audio and visual features in the temporal dimension? ❸ In the open-vocabulary setting, the model is trained on a closed set and then localizes while identifying sounding objects within categories beyond the annotated label space. This calls for the model to possess strong generalization abilities: not only localizing sounding objects, but also suppressing silent objects and background disturbances, such as noise or sounds from objects outside the frame.

To address the above problems, we design the following modules. For problem ❶, an audio-visual early fusion module is introduced to align audio signals with image features in the spatial dimension. It takes an image feature map from the visual backbone and the corresponding audio embedding from the pre-trained audio encoder as inputs, and then computes bi-directional cross-attention between them. On the one hand, the audio embedding as the query can be enhanced by correlating with image pixels of those semantic-relevant objects. On the other hand, the image feature as the query can be supplemented and activated with audio information. For problem ❷, we present an audio-conditioned Transformer decoder to establish frame-by-frame relationships and capture audio-visual dependencies in the temporal dimension. Specifically, a set of learnable object queries is first fed into the decoder and performs the cross- and self-attention sequentially to obtain object-centric representations, following Mask2Former [2]. Then, these queries extract audio temporal information via a cross-attention layer, where the key elements are the all audio embeddings from the audio-visual early fusion module. For problem ❸, to localize sounding objects from all categories, we discard the class head but added an sound head which acts as a binary classifier only responsible for determining

whether the object is making sound in the current frame. This enables our model to not be constrained by the close-set classification annotations. Additionally, we employ the large-scale pre-trained vision-language model, CLIP [26], to predict the categories of the above sounding objects.

On the whole, we propose an open-vocabulary audio-visual semantic segmentation model, termed OV-AVSS, which consists of two main components: universal sound source localization module (USSLM; detailed in Section 3.3) and open-vocabulary classification module (OVCM; detailed in Section 3.4). Equipped with the audio-visual early fusion module and audio-conditioned Transformer decoder, USSLM integrates audio and visual features in spatial and temporal domains, separately, and then outputs class-agnostic sounding objects. With the aid of the rich knowledge from CLIP, OVCM classifies these potential sounding objects without limiting to closed-label space. To assess the model's generalization capability on unseen and unheard objects, we build a open-vocabulary audible video dataset based on AVSBench-Semantic benchmark [36]. Experimental results demonstrate that OV-AVSS achieves superior generalizable segmentation performance (55.43% mIoU on base categories and 29.14% mIoU on novel categories).

To sum up, our contributions are threefold:

(1) We propose a new multi-modal task, open-vocabulary audio-visual semantic segmentation, which aims at segmenting and classifying sounding objects of arbitrary open-set categories in videos.

(2) A strong baseline model is developed to generate class-agnostic masks for sounding objects and predict their categories by leveraging the prior knowledge of large-scale vision-language models.

(3) Extensive experiments indicate that our model can achieve state-of-the-art results in terms of performance and generalization. It has been ready-to-use for open-vocabulary audio-visual semantic segmentation in the real-world applications.

## 2 RELATED WORKS

### 2.1 Audio-Visual Segmentation

Audio-visual segmentation (AVS) aims to segment the objects that emit sound within each video frame. Recent AVS research primarily focuses on supervised learning on the sparsely annotated AVSBench dataset, which can be divided into FCN-based and Transformer-based approaches. For FCN-based methods, Zhou *et al.* [37] propose temporal pixel-wise audio-visual interaction (TPAVI), which is a non-local block with cross-modal attention to locate the sound source. ECMVAE [24] employs three latent encoders to achieve latent space factorization, yielding both modality-shared and specific representations to model audio-visual contributions explicitly. In addition, Transformer-based methods accomplish audio-visual fusion and mask decoding with vision Transformer [6]. For example, AVSegFormer [7] uses audio-based channel attention to dynamically adjust visual features, and adopts a DETR-like architecture [1] to segment sounding objects. Liu *et al.* [19] associate the audio signals with visual information by aligning predicted audio class labels with instance segmentation masks, thereby highlighting sounding objects while suppressing silent ones. Audio-visual semantic segmentation (AVSS) extends the AVS task by additionally predicting the category of each sound source. Due to the semantic entanglement in audio, tackling multi-source AVSS presents a

greater challenge. Zhou *et al.* [36] follow the TPAVI module [37] to perform audio-visual fusion, and then attach a class head for semantic predictions. CATR [17] separately carries out audio-visual interactions in a decoupled manner. Additionally, audio-constrained queries are introduced to select sounding objects during decoding.

## 2.2 Zero-Shot Audio-Viusal Segmentation

Current AVS approaches mainly focus on in-domain and close-set situations. When faced with unseen classes in real-world scenarios, these approaches will no longer be applicable, owing to the limited training data. To overcome the drawback, Wang *et al.* [28] design an encoder-prompt-decoder framework to improve the generalization of the AVS model by leveraging the prior knowledge of the visual foundation model. Specifically, they use the adapter technique [14] to fine-tune the pre-trained segment-anything model (SAM) [15] and generalize knowledge learned from the training set to unseen objects. Yu *et al.* [33] employ the large-scale audio-language model, CLAP [30], to predict the category to which the audio belongs. For the single-source and multi-source audio, they select the class names with the highest and the top two highest scores, respectively. The predicted class names are input into Grounding DINO [20] and SAM [15] to sequentially generate box predictions and the corresponding masks. Unlike the above zero-shot methods applied to the binary segmentation, this work focuses on the audio-visual semantic segmentation and exploits the large-scale pre-trained CLIP model [26] to classify sounding objects from novel categories.

## 2.3 Open-Vocabulary Segmentation

Compared with traditional segmentation methods [11, 35] that only identify pre-defined categories in the training set, open-vocabulary segmentation seeks to locate and recognize categories beyond the annotated label space. The main difference between zero-shot learning and open-vocabulary learning is that the latter is capable of leveraging visual-related language vocabulary data [29]. For example, LSeg [16] aligns the image embeddings with the text embeddings of category labels from CLIP text encoder. This allows LSeg to use the knowledge of vision-language models and segment objects that are not pre-defined but depend on the input texts. ZegFormer [5] decouples the problem into a class-agnostic segmentation task and a zero-shot classification task. It uses the CLIP image encoder to obtain pixel embeddings of masked images and then utilizes label embeddings from CLIP text encoder to classify the proposal masks. Recently, more works focus on video instance segmentation and attempt to build an open-vocabulary tracker based on close-set training data. Wang *et al.* [27] collect a large-scale video dataset, and propose an end-to-end approach, called OV2Seg, for open-vocabulary video instance segmentation. OV2Seg adopts a momentum-updated module to track objects and a CLIP-based classifier for recognizing novel categories. OpenVIS [10] adapts the pre-trained CLIP with instance guidance attention and generates object proposals and the corresponding classes. Moreover, a rollout association mechanism is designed to associate instances of any categories across frames thereby improving tracking performance. In this paper, we extend open-vocabulary learning to audio-visual semantic segmentation: segmenting and classifying sound-producing objects of arbitrary open-set categories simultaneously.

## 3 METHOD

In this section, we present an open-vocabulary audio-visual semantic segmentation framework, namely OV-AVSS, as shown in Fig. 2. In the first stage, the universal sound source localization module (Section 3.3) is applied to predict class-agnostic masks of sounding objects covering all possible classes. In the second stage, the open-vocabulary classifier (Section 3.4) empowered by CLIP [26], is employed to identify the category of each sounding object mask.

## 3.1 Problem Definition

Let $D_{train}$ be a training dataset containing pixel-level annotations for a set of base categories $C_{base}$, open-vocabulary audio-visual semantic segmentation aims to train a model $f_\theta(\cdot)$ on $D_{train}$, and then test on $D_{test}$ for both base categories $C_{base}$ and novel categories $C_{novel}$ with the help of large language vocabulary knowledge $C_{voc}$. Note that $C_{voc}$ is not strictly required to contain $C_{base}$ or $C_{novel}$, as the language vocabulary may not cover all the class names in the vision or audio data. Given a test video clip $\mathbf{V} \in \mathbb{R}^{T \times H \times W \times 3}$ and the corresponding audio $\mathbf{A} \in \mathbb{R}^T$, $f_\theta(\cdot)$ is supposed to predict the segmentation mask $\{\mathbf{m}_i\}_{i=1}^T \in \mathbb{R}^{T \times H \times W}$ and the category label $\mathbf{c} \in (C_{base} \cup C_{novel})$ for each sounding object of all the frames:

$$\{\{\mathbf{m}_1, \mathbf{m}_2, \ldots, \mathbf{m}_T\}, \mathbf{c}\}_P^P = f_\theta(\mathbf{V}, \mathbf{A}),$$

where $P$ is the total number of categories of the sounding objects. In the experiments, we designate the categories used for training as "base categories". Categories that fall outside the established base categories are referred to as novel categories.

## 3.2 Multi-modal Representation

Given an input video sequence containing both visual and audio tracks, we divide it into $T$ non-overlapping audio and visual snippet pairs $\{\mathbf{V}, \mathbf{A}\} = \{\mathbf{v}_i, \mathbf{a}_i\}_i^T$, where each snippet is 1-second long and T denotes the number of snippets as well as the video duration.

**Audio Representation.** For each audio snippet $\mathbf{a}_i$, we use the pre-trained VGGish [8] model to extract the audio feature as $\mathbf{f}_i^A \in \mathbb{R}^C$, where $C = 128$ is the feature dimension. The VGGish model, a VGG-like 2D convolutional neural network, is pre-trained on AudioSet dataset [8]. Due to the input being 2D audio spectorgram, we first convert the audio $\mathbf{A}$ into a mono-waveform at a sampling rate of 16kHz and apply short-time Fourier transform to obtain the mel spectrograms. Then, they are fed into VGGish model and the audio features are extracted as $\mathbf{F}^A = \{\mathbf{f}_1^A, \mathbf{f}_2^A, ..., \mathbf{f}_T^A\}$. Note that the audio representation is extracted offline and the parameters of VGGish model are frozen during the training process.

**Visual Representation.** For each visual snippet $\mathbf{v}_i$, we extract image-level hierarchical features using the convolutional-based ResNet-50 [13] or Transformer-based Swin-B [22] backbones. We represent the features as $\mathbf{f}_{i,k}^V \in \mathbb{R}^{H_k \times W_k \times D_k}$, where $(H_k, W_k) = (H, W)/2^{k+1}$, $k = 1, 2, 3, 4$, $(H, W)$ is the resolution of each visual snippet $\mathbf{v}_i$ and $k$ is the number of backbone levels. The final visual representation can be formulated as $\mathbf{F}^V = \{\mathbf{f}_1^V, \mathbf{f}_2^V, ..., \mathbf{f}_T^V\}$.

## 3.3 Universal Sound Source Localization

To enhance the audio features $\mathbf{F}^A$ with the image contexts and embed audio information into image features $\mathbf{F}^V$ simultaneously,

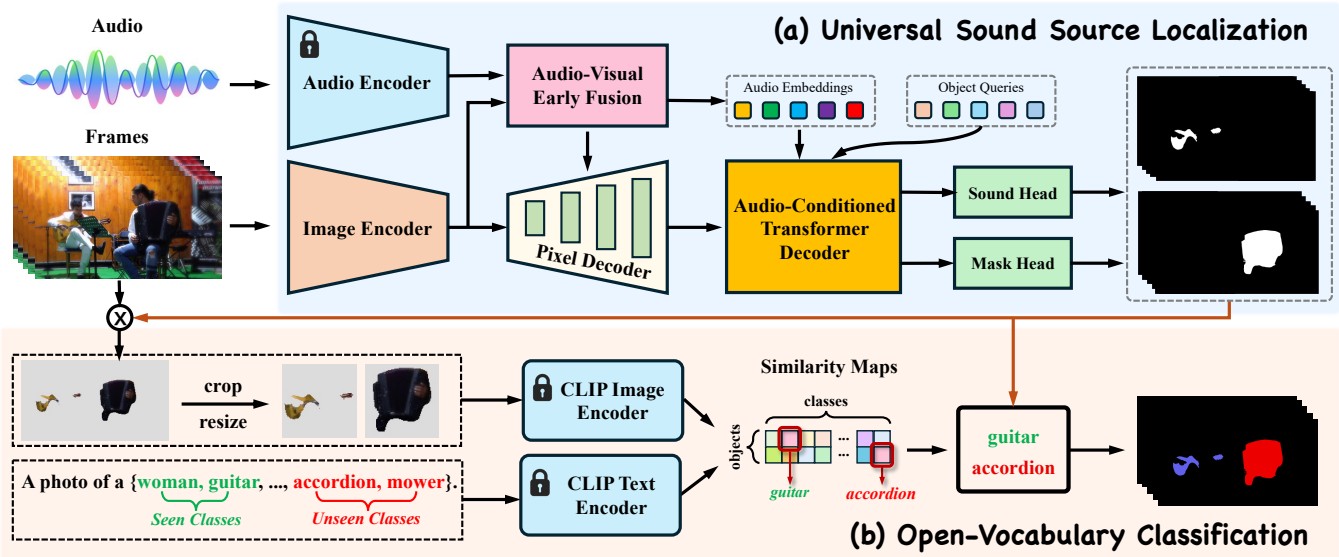

**Figure 2: Overview of the proposed OV-AVSS.** (a) Universal Sound Source Localization: **Given the image and audio features, the audio-visual early fusion module takes them as input and aligns them in spatial domain. Then, the fused features are passed into the pixel decoder and audio-conditioned Transformer decoder, which captures audio-visual dependencies in temporal domain and generates the class-agnostic mask for each sounding object.** (b) Open-Vocabulary Classification: **After localizing sounding objects and obtaining their masks, we crop the input frames with masks and feed into CLIP image encoder to generate image embeddings. They are then dot-producted with text embeddings generated by CLIP text encoder to obtain object categories.**

an audio-visual early fusion module is adopted, allowing modality interaction at the early stage (before feature decoding). Specifically, the audio feature $\mathbf{f}_i^A$ and multi-level image features $\{\mathbf{f}_{i,2}^V, \mathbf{f}_{i,3}^V, \mathbf{f}_{i,4}^V\}$ are linearly projected to the same dimension $C_{av}$ via pointwise convolution and group normalization. Then, we flatten image features and concatenate them into a 1D sequence $\mathbf{s}_i^V$. Finally, a bi-directional cross-attention module ($BC_{A \to V}$ and $BC_{V \to A}$) is applied to establish audio-visual spatial dependence and obtain cross-modal feature embeddings $\mathbf{f}_i^{AV}, \mathbf{f}_i^{VA}$. This process can be formulated as:

$$\mathbf{f}_i^{AV} = BC_{A \to V}(\mathbf{f}_i^A, \mathbf{s}_i^V) = \text{Softmax}\left(\frac{\mathbf{f}_i^A \mathbf{W}^Q \cdot (\mathbf{s}_i^V \mathbf{W}^K)^T}{\sqrt{d_k}}\right) \cdot \mathbf{s}_i^V \mathbf{W}^V + \mathbf{f}_i^A \quad (1)$$

$$\mathbf{f}_i^{VA} = BC_{V \to A}(\mathbf{s}_i^V, \mathbf{f}_i^A) = \text{Softmax}\left(\frac{\mathbf{s}_i^V \mathbf{W}^Q \cdot (\mathbf{f}_i^A \mathbf{W}^K)^T}{\sqrt{d_k}}\right) \cdot \mathbf{f}_i^A \mathbf{W}^V + \mathbf{s}_i^V \quad (2)$$

where $\mathbf{W}^Q, \mathbf{W}^K, \mathbf{W}^V \in \mathbb{R}^{C_{av} \times d_k}$ are learnable parameter matrices. In $BC_{A \to V}$, it serves audio feature $\mathbf{f}_i^A$ as query and 1D image feature $\mathbf{s}_i^V$ as key/value, and then perform multi-head attention between them. Owing to the image feature $\mathbf{s}_i^V$ being multi-level and processed only once, the audio-visual early fusion module can efficiently leverage both strong semantics and fine-grained details. In $BC_{V \to A}$, query is $\mathbf{s}_i^V$ and key/value is $\mathbf{f}_i^A$. It treats the audio signals as auxiliary information, and the embedded semantical consistency is used to highlight the corresponding spatial regions.

Given the processed image features $\mathbf{f}_i^{VA}$, we feed them into the pixel decoder, a multi-scale deformable attention Transformer [40], to output the enhanced feature $\mathbf{o}_i^{VA}$ and high-resolution per-pixel embeddings $\mathbf{p}_i$. Considering sound source localization requires

learning more accurate audio-object matching relationship, we propose the audio-conditioned Transformer decoder (AudioMaskDec) to incorporate audio information and decode the masks for sound-emitting objects. As shown in Fig. 3, AudioMaskDec consists of five key components: a spatio-temporal cross-attention layer, an audio self-attention layer, an object self-attention layer, an audio-aware cross-attention layer, and a feedforward neural network.

To be specific, we first initialize a set of learnable object queries $\mathbf{q} \in \mathbb{R}^{N \times C_o}$ supplemented with positional encodings, where $N$ is the number of object queries. Then, the spatio-temporal cross-attention layer computes cross-attention between object queries and all frames' features. This empowers object queries $\mathbf{q}'$ with visual semantics and associates them with all potential objects, following previous works [2, 3]. After that, the object queries $\mathbf{q}'$ and audio embeddings $\{\mathbf{f}_i^{AV}\}_{i=1}^T$ are fed into the object self-attention layer and audio self-attention layer, separately. With the help of two separated self-attention layers, it isolates $\mathbf{q}'$ and $\{\mathbf{f}_i^{AV}\}_{i=1}^T$ instead of concatenating together as inputs, so as to facilitate object interactions across spatial domains and audio interactions across time domains, respectively. We denote the processed object queries as $\mathbf{q}^*$ and audio embedding as $\{\mathbf{f}_i^*\}_{i=1}^T$. To better judge when objects emit sound, we introduce an audio-aware cross-attention layer that performs cross-attention between $\mathbf{q}^*$ and $\{\mathbf{f}_i^*\}_{i=1}^T$, where the former is query and the latter is key/value. In this way, T-second audio is embedded into object queries to guide sound source localization temporally. Inspired by [18], the audio-aware cross-attention layer is placed behind spatio-temporal cross-attention layer to avoid forgetting audio information as the decoder layer goes deep.

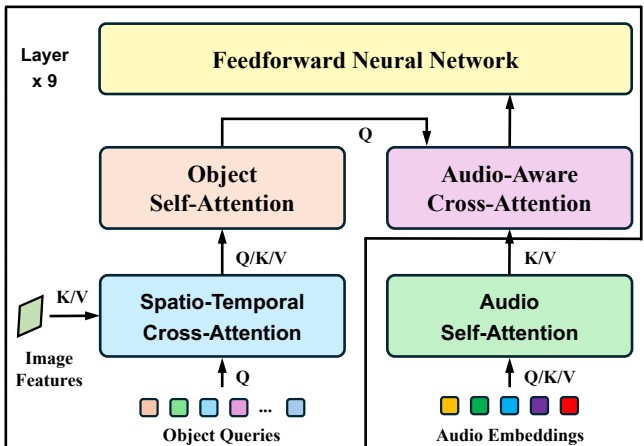

**Figure 3: The architecture of our proposed audio-conditioned Transformer decoder.** $N$ **class-independent object queries learn semantics from image features and capture audio-visual temporal dependencies from audio embeddings.**

To detect the sounding objects from all categories while ignoring other non-sounding objects, a sound head is attached to the AudioMaskDec, which predicts the binary sounding scores $S \in \mathbb{R}^{N \times 1}$, indicating if a query represents an sounding object:

$$S = \text{Softmax}(\text{MLP}_{sound}(\text{AudioMaskDec}(q))) \qquad (3)$$

Due to the utilization of the sound head instead of a class head, object queries are class-agnostic, enabling our model to detect objects from all categories. This universal object proposal has been widely proved in previous open-vocabulary works [27, 34]. For the mask prediction, the class-agnostic object queries are passed into a mask head and then dot-produced with the per-pixel embeddings $p_i$, yielding the final masks $\{M_i\}_{i=1}^{T}$.

$$M_i = \text{MLP}_{mask}(\text{AudioMaskDec}(q)) \circledast p_i, \forall i \in [1, T] \qquad (4)$$

## 3.4 Open-Vocabulary Classification

Once sounding object candidates are localized, we employ the frozen pre-trained vision-language model, i.e., CLIP [26], to classify each candidate. Concretely, the category names with prompt templates are input into the CLIP text encoder $\text{CLIP}_{text}$ to generate the text embeddings offline, such as:

$$E_{text} = \text{CLIP}_{text}(\text{"This is a photo of a \{violin\}"}) \qquad (5)$$

where the category names contain both base and novel classes. Since CLIP model is trained on full sentences, we ensemble multiple prompt templates to improve classification accuracy. Our list of prompt templates is shown in Appendix.

In addition, we crop sounding image regions with the proposed masks $M_i^n$ and feed into the CLIP image encoder $\text{CLIP}_{image}$ to compute image embeddings $E_{image}$, where $n$ represents the number of sounding objects. Inspired by [9, 10], we first overlay the mask $M_i^n$ onto the original image, resulting in the masked images. Then, we compute the object's center coordinate and the length of its longer side. Next, the masked images are cropped into a square-shaped images $I_i^n$. Lastly, we resize $I_i^n$ to $224 \times 224$ resolution required

by the CLIP image encoder, such as ViT-B [6]. Compared with no cropping or simple cropping in [32], this square cropping strategy can avoid objects occupying too small an area of the image and ensure $I_i^n$ without distortion.

After acquiring the text embeddings $E_{text} \in \mathbb{R}^{cls \times D_c}$ and image embeddings $E_{image} \in \mathbb{R}^{n \times D_c}$, we compute the similarity matching map $Sim \in \mathbb{R}^{n \times cls}$ between them as:

$$Sim(i, j) = \text{softmax}(E_{image}{}^i \cdot (E_{text}{}^j)^T \cdot \epsilon) \qquad (6)$$

where $cls$ is the number of all categories, including base and novel classes. $\epsilon$ is the temperature hyper-parameter. The class label of each class-agnostic mask can be obtained by the argmax operation.

## 3.5 Training and Loss

In the training phase, suppose there are $K$ sounding objects in an image, we select $K(K < N)$ object queries that best refers to the sounding objects via a bipartite matching strategy. Technologically, we use the sounding scores and mask predictions to compute the assignment cost metrics. The cost metrics is composed of two parts: one for segmentation that includes binary focal loss and dice loss, and the other is binary classification for sounding objects presence. Then the $K$ best matched object queries $\{\hat{q}_k\}_{k=1}^{K}$ are applied for model training. Given the above matching, each prediction is supervised with a sound score loss and a mask loss. The former is binary cross-entropy loss and the latter consists of a focal loss and a dice loss. The total loss function between the query $\{\hat{q}_k\}_{k=1}^{K}$ and ground-truth $\{y_k\}_{k=1}^{K}$ can be written as:

$$\mathcal{L}(\hat{q}_k, y_k) = \lambda_{ce}\mathcal{L}_{ce} + \lambda_{focal}\mathcal{L}_{focal} + \lambda_{dice}\mathcal{L}_{dice} \qquad (7)$$

where $\lambda_{ce} = 2.0$, $\lambda_{focal} = 5.0$, and $\lambda_{dice} = 5.0$ are the weights to balance the loss function.

## 4 EXPERIMENTS

## 4.1 Datasets and Evaluation Metrics

To facilitate a comprehensive evaluation of OV-AVSS, we partition a novel dataset called **AVSBench-OV**, derived from AVSBench-Semantic [36], an extension of the initial AVSBench-object dataset [37] for semantic segmentation. AVSBench-Semantic encompasses 70 categories, incorporating both single and multiple sound source scenarios. Each video within AVSBench-Semantic is truncated to either 5 or 10 seconds in duration, with one frame extracted per second. Following the partitions in LVIS [12], we split the categories within AVSBench-OV into 40 base categories representing those seen during training and inherited from frequent and common categories, and 30 novel (unseen and unheard) categories disjoint from the base categories. To ensure consistency, we eliminated sample videos from the training subset whenever a novel category appeared in annotations, thereby restricting the training data exclusively to base categories. The dataset statistics of AVSBench-OV are reported as follows: the training set encompasses 5,184 real-world videos comprising 40,095 frames across 40 base categories, while the validation and test sets consist of 1,240 and 1,490 videos across all 70 categories. For specific category names and corresponding video counts per category, please refer to the Appendix.

Following the previous open-vocabulary semantic segmentation task [25, 31], we adopt the mean intersection-over-union ($mIoU$)

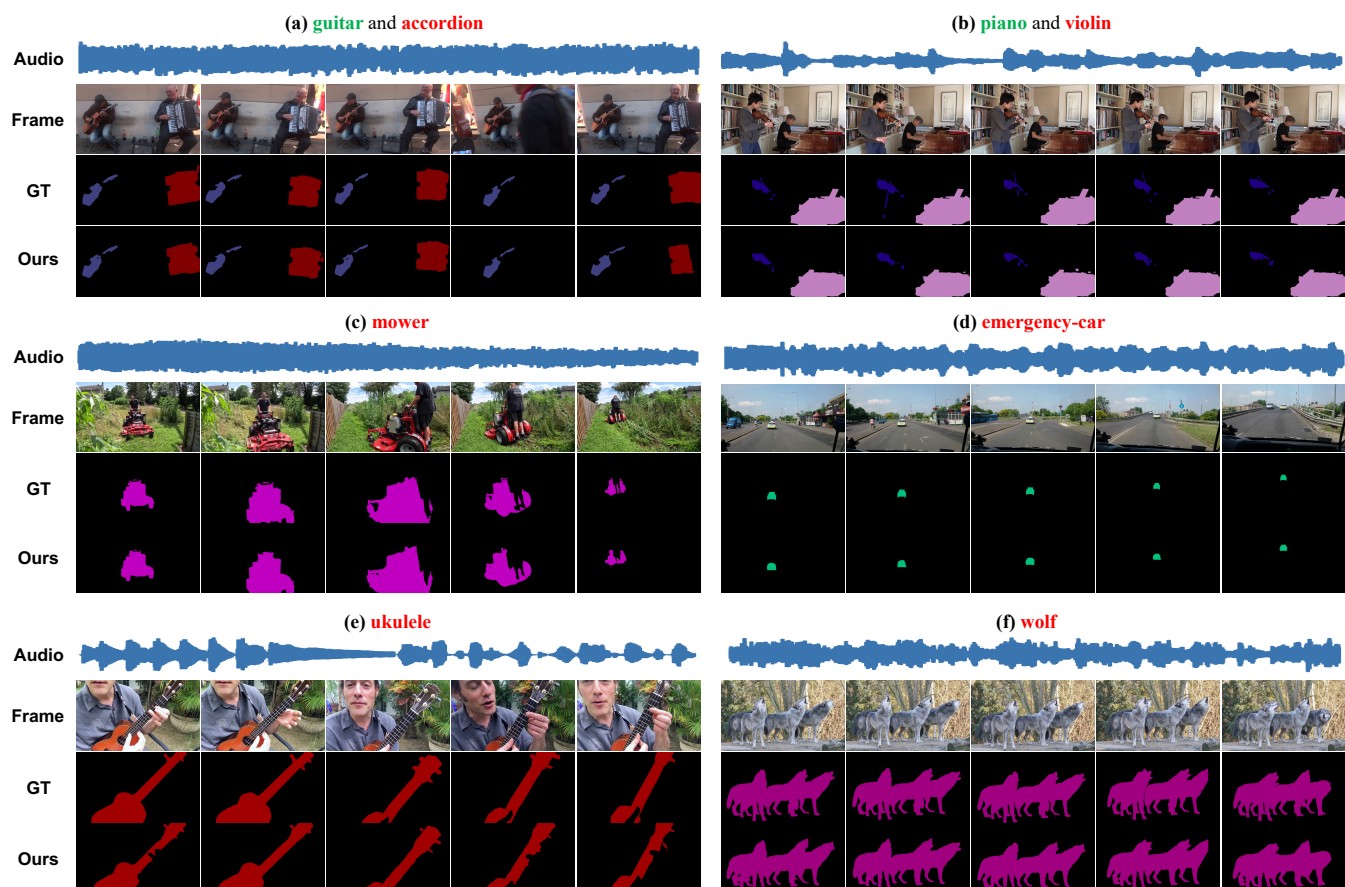

**Figure 4: Qualitative results of our novel OV-AVSS framework on diverse audio-visual scenarios. Categories of predictions are shown in the title. Green text represents base categories, while red text denotes novel categories. (a) and (b) are multi-source scenarios containing both base and novel categories. (c)-(e) present single-source scenarios with one novel category. (f) is multi-source scenario with one novel category. Best view in color.**

to evaluate the open-vocabulary AVSS performance. We compute overall mIoU, Base mIoU for seen categories, Novel mIoU for unseen categories, along with their harmonic mean (Harmonic). Harmonic is defined as follows:

$$Harmonic = \frac{2 * Base * Novel}{Base + Novel} \quad (8)$$

## 4.2 Implementation Details

We use both ResNet-50 [13] and Swin-B [22] as the image encoders and VGGish [8] as the audio encoder. Parameters in the audio encoder, CLIP image encoder and CLIP text encoder are frozen during the training. For each video clip, we set the total number of video frames $T$ to 5 by default. The number of object queries is set to 100 for all experiments. We train our model on the AVSBench-OV dataset for 82,000 iterations (about 10 epochs) with a batch size of 1. We use the AdamW optimizer [23] and the step learning rate schedule. The initial learning rate is 1e-4 and scaled by a decay factor of 0.1 at the 72,171 iterations. If no other specification, the shorter side of frames are resized to 240, 360 or 480 during training. The experiments are conducted on an NVIDIA Quadro 6000 GPU.

## 4.3 Comparison to State-of-the-art Methods

**Table 1: Comparison with other zero-shot audio-visual segmentation methods on AVSBench-OV dataset. The "Base", "Novel", and "Harmonic" are mIoU of base classes, novel classes, and their harmonic mean. "AVBS" and "AVSS" denote audio-visual binary and semantic segmentation, respectively.**

| Model | Task | Base | Novel | Harmonic | mIoU |
|-------|------|------|-------|----------|------|
| GAVS [28] | AVBS | - | - | - | - |
| Sam4AVS [33] | AVSS | 13.55 | 8.53 | 10.47 | 12.47 |
| **OV-AVSS** | AVSS | **55.43** | **29.14** | **38.20** | **44.81** |

**Zero-Shot Audio-Visual Segmentation.** Given that our proposed open-vocabulary AVSS task employs a new division of base and novel categories, there are no baselines with identical training settings for an entirely fair comparison. Therefore, we use baselines from a similar but not identical tasks, specifically, the zero-shot AVS

**Table 2: Comparison of AVSS performance on AVSBench-OV between closed-set and open-vocabulary methods.**

| Type | Model | Pub. & Year | Audio | Vision Backbone | Text Encoder | Base | Novel | Harmonic | mIoU |
|------|-------|-------------|-------|-----------------|--------------|------|-------|----------|------|
| Closed-Set | TPAVI [37] | ECCV 22 | ✔ | ResNet-50 | *none* | 28.45 | 0.00 | 0.00 | 18.08 |
| | TPAVI [37] | ECCV 22 | ✔ | PVT-v2 | *none* | 33.25 | 0.00 | 0.00 | 22.02 |
| | CATR [17] | MM 23 | ✔ | ResNet-50 | *none* | 30.64 | 0.00 | 0.00 | 19.73 |
| Open-Vocabulary | Detic[38]-XMem[4] | ECCV 22 | ✘ | ResNet-50 | CLIP | 22.37 | 8.30 | 14.10 | 22.88 |
| | OV2Seg [27] | ICCV 23 | ✘ | ResNet-50 | CLIP | 36.26 | 12.49 | 18.58 | 26.93 |
| | OpenVIS [10] | ArXiv 23 | ✘ | ResNet-50 | CLIP-ViT-B/32 | 45.23 | 17.54 | 25.31 | 34.40 |
| | CLIP-VIS [39] | ArXiv 24 | ✘ | ResNet-50 | CLIP | 40.35 | 15.00 | 21.87 | 30.31 |
| | **OV-AVSS** | - | ✔ | ResNet-50 | CLIP-ViT-B/16 | **49.77** | **22.20** | **30.71** | **38.91** |
| | **OV-AVSS** | - | ✔ | Swin-base | CLIP-ViT-L/14 | **55.43** | **29.14** | **38.20** | **44.81** |

task. First, we consider GAVS [28], which introduces an encoder-prompt-decoder paradigm to enhance the generalization of the AVS model by leveraging the prior knowledge of SAM [15]. However, due to the absence of CLIP or alternative foundation models, GAVS is limited to generate binary masks and cannot be applied to AVSS. Additionally, we compare with Sam4AVS [33], which first employs CLAP [30] to predict categories of the input audio, and then feed into Grounding DINO [21] and SAM for mask prediction. A significant constraint of Sam4AVS stems from its dependency on the CLAP for class identification, which neglects the visual cues. This exclusion can result in inaccuracy classification for sounding objects. Also, Sam4AVS segments all instances of the predicted category, lacking the ability to identify specific sounding source. The original Sam4AVS is used for binary segmentation. We reproduce it and make it suitable for the open-vocabulary AVSS task. As shown in Table 1, our OV-AVSS model surpasses Sam4AVS in all metrics by a significant margin (+41.88% mIoU in base categories, +20.61% mIoU in novel categories, +27.73% harmonic mIoU and +32.34% overall mIoU). These substantial improvements can be attributed to our designed universal sound source localization and open-vocabulary classification modules, which effectively leverages both audio and visual information for accurate segmentation and categorization of both seen and unseen sounding objects.

**Open-Vocabulary Audio-Visual Semantic Segmentation.** As shown in Table 2, we first compare OV-AVSS with two closed-set trained methods, namely TPAVI [37] and CATR [17]. These methods adhere to the traditional paradigm based on the close-set assumption, which can only predict pre-defined categories that are present in the training set. Consequently, their performance on the novel set yields zero across all metrics. To provide a thorough comparison, we evaluate TPAVI with different backbones, including ResNet-50 and PVT-v2. The results demonstrate that even on the base categories, the performance of these closed-set methods falls short of our approach. Furthermore, we compare OV-AVSS to several state-of-the-art open-vocabulary video instance segmentation methods. These methods do not incorporate audio cues, making it challenging to localize sounding objects in frames. Our proposed OV-AVSS significantly outperforms the best-performing method, CLIP-VIS [39], by +9.772% in base categories, +7.2% in novel categories, +8.84% in harmonic mIoU, and +8.6% in overall mIoU. These results highlight the effectiveness of our approach in leveraging

both visual and audio information to achieve superior performance in open-vocabulary audio-visual semantic segmentation.

**Qualitative Results.** In Fig. 4, we present qualitative semantic segmentation results of the proposed OV-AVSS model on both base and novel categories. The diverse range of scenarios presented in Fig. 5, including machines, musical instruments, and animals, demonstrates the broad applicability of OV-AVSS. Our model can precisely localize the sounding object within each frame, e.g. emergency car in Fig. 4 (d), while effectively omitting similar objects (vehicles) that do not emit sound. The multi-source scenarios depicted in Fig. 4 (a) and (b) show our method's remarkable ability to accurately segment sounding objects not only from base categories but also from unseen categories, i.e. "accordion" and "violin". This highlights the effectiveness of our approach in handling complex scenes containing a mix of seen and unseen sound sources. Despite the prevalence of "human" as the most easily confused sounding object in the training set, our method also accurately isolates silent humans, as evidenced by the examples shown in Fig. 4 (a), (b), and (e). Moreover, in the single-source scenario depicted in Fig. 4 (e), our segmentation for the novel category "ukulele" exhibit higher precision than the ground truth, i.e. fingers in $1^{st}$ & $4^{th}$ frame. This underscores our framework's capacity to leverage audio-visual cues for accurate semantic segmentation, even surpassing human annotations in certain cases. Fig. 4 (f) illustrates OV-AVSS's ability to segment multiple instances of the same novel category, "wolf," within a single scene. This showcases the robustness of our framework in handling multi-sounding objects of the same category.

### 4.4 Ablation Study

**Table 3: Impact of Audio-Visual Early Fusion Strategy.**

| Fusion Method | Multi-level | Base | Novel | Harmonic | mIoU |
|---------------|-------------|------|-------|----------|------|
| None | ✘ | 47.38 | 20.69 | 28.80 | 36.89 |
| Add | ✔ | 48.94 | 21.73 | 30.10 | 38.24 |
| Bi-Attn | ✘ | 49.22 | 21.89 | 30.30 | 38.46 |
| Bi-Attn | ✔ | **49.77** | **22.20** | **30.71** | **38.91** |

**Impact of Audio-Visual Early Fusion Strategy.** The setting on the first row of Table 3 serves as a baseline, where no early

fusion is applied. While this baseline approach can generate reasonably good results for both base and novel categories, there is still room for improvement. The audio embeddings and processed image features, when used independently, may not capture sufficient semantic information to guide the localization process effectively. To address this limitation, we introduce an early fusion mechanism that allows us to acquire audio-aligned visual features and visual-enriched audio embeddings. The second row in Table 3 presents a straightforward fusion approach, where audio embeddings are broadcast and added to the visual features in a pixel-wise manner. However, it is important to note that this fusion strategy is a one-way process, where the audio embeddings are used to enhance the visual features, but the audio embeddings themselves do not receive any visual priors to aid in sound source localization. To further improve the fusion mechanism, we introduce a bi-directional cross-attention mechanism, where each audio embedding can acquire corresponding semantic information from the visual features. Moreover, the cross-attention operation is much stronger than that of simple one-way summation, as it can adaptively focus on relevant regions in the visual features based on the audio embeddings. This brings extra performance increase as shown in the $4^{th}$ row of Table 3, with improvements of +0.83% in base, +0.47% in novel, +0.61% harmonic mean, and +0.73% overall mIoU. This process is performed at multiple levels, allowing the model to incorporate audio information at different scales.

**Table 4: Impact of Audio Prompt Method.**

| Audio Prompt Method | Base | Novel | Harmonic | mIoU |
|---|---|---|---|---|
| None | 46.77 | 19.98 | 27.99 | 36.23 |
| Concat+Add | 48.02 | 20.38 | 28.62 | 37.12 |
| Cross-Attn | 49.19 | 20.79 | 29.22 | 37.95 |
| AudioMaskDec | **49.77** | **22.20** | **30.71** | **38.91** |

**Impact of Audio Prompt Method.** For the baseline, we directly feed the object queries into the decoder, disregarding the audio embeddings obtained through our audio-visual early fusion. Due to the pixel-wise audio information incorporated into the image features through early fusion, this approach achieves promising results. However, it fails to fully utilize the temporal audio information, limiting its potential for enhancing performance. To overcome this obstacle, we concatenate the audio embeddings in the temporal dimension and add them to the object queries, similar to positional encoding. While this method yields modest improvements across all categories (+1.25% in base, +0.4% in novel, +0.63% harmonic, and +0.89% mIoU), it does not fully capture the rich time-dependent correspondence present in the audio modality. We further examine the impact of performing a single cross-attention operation between the object queries and audio embeddings before feeding them into the decoder. This approach yields further improvements in performance, highlighting the importance of effectively integrating audio and visual information. However, compared to our proposed AudioMaskDec, this audio prompting method may suffer from information loss as the decoder depth increases, limiting its ability to fully exploit the audio-visual synergy. In contrast, AudioMaskDec

places an audio-aware cross-attention layer behind each spatio-temporal cross-attention layer in the decoder. This design enables the model to iteratively refine the integration of audio and visual features throughout the decoding process. By facilitating the interaction between object queries and audio embeddings at multiple layers, AudioMaskDec ensures that pertinent audio information is retained and effectively utilized before the sound head and mask head. As shown by the results in row 4 of Table 4, this approach further enhances the model's performance by +0.58% Base, +1.41% Novel, +1.49% Harmonic, and +0.96% overall mIoU.

**Table 5: Impact of Image Crop Strategy.**

| Crop Strategy | CLIP Encoder | Base | Novel | Harmonic | mIoU |
|---|---|---|---|---|---|
| None | ViT-B | 46.23 | 18.43 | 26.36 | 35.32 |
| CropResize | ViT-B | 47.61 | 20.92 | 29.06 | 37.14 |
| SquareCrop | Res-50 | **50.15** | 21.80 | 30.39 | 38.71 |
| SquareCrop | ViT-B | 49.77 | **22.20** | **30.71** | **38.91** |

**Impact of Image Crop Strategy.** Previous work on zero-shot semantic segmentation [31] has shown that cropping an object's masked region by using its bounding box and resizing to a fixed size (e.g. $224 \times 224$) provides a more suitable input representation for CLIP compared to the full image. However, naively resizing the masked region can introduce significant aspect ratio distortion relative to the object's natural proportions. This unnatural stretching may degrade CLIP's zero-shot classification performance. We instead compute the center coordinate and longer side length of the object's bounding box, and extract a square crop around the object center, yielding a distortion-free representation. As shown in Table 5 row 2 and 4, SquareCrop approach outperforms the CropResize by a sizeable margin across all metrics, including a +2.16% gain on base classes, +1.28% on novel classes, +1.65% on harmonic mean, and +1.77% on mIoU. Furthermore, we investigate the impact of using different visual encoders of CLIP. As shown in the $3^{rd}$ row of Table 5, changing the visual encoder does not yield further improvements in performance. This suggests that the classification effect of CLIP is largely dependent on the segmentation quality of the universal module and the input image quality, rather than the choice of different visual encoder.

## 5 CONCLUSION

This paper introduces a new task of open-vocabulary AVSS with the goal of segmenting and classifying sounding objects of arbitrary categories in a given video. Furthermore, we present the first open-vocabulary AVSS framework, termed OV-AVSS, generalizing from annotated (seen) object classes to other (unseen) categories with the knowledge of CLIP. Quantitative and qualitative experiments on AVSBench-OV dataset show strong zero-shot generalization ability on novel categories unseen during training. We hope that our proposed OV-AVSS model can further promote the generalization study of audio-visual segmentation in zero-shot, open-vocabulary and real-world scenarios. In the future, we will consider using extra video caption data to guide segmentation, and extend the model to class-aware audio-visual detection.

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
