# OpenReview forum: "Open-Vocabulary Audio-Visual Semantic Segmentation"
_acmmm.org/ACMMM/2024/Conference — MM2024 Oral_

### Official Review · Reviewer_2fEx · 2024-05-06

**Rating:** 4
**Confidence:** 1

**Summary:**

The paper introduces a new task called open-vocabulary audio-visual semantic segmentation (OV-AVSS), which aims to segment and classify sounding objects in videos from open-set categories. The authors propose a novel framework, OV-AVSS, which consists of two modules: a universal sound source localization module and an open-vocabulary classification module. The framework integrates audio and visual features in spatial and temporal domains and leverages prior knowledge from large-scale vision-language models to predict categories. Experimental results on a new dataset, AVSBench-OV, demonstrate the strong segmentation and zero-shot generalization ability of the model.

**Strengths:**

The paper introduces a new and challenging task, OV-AVSS, which has practical applications in real-world scenarios.
The proposed framework, OV-AVSS, is a novel and effective approach to tackle the task, achieving state-of-the-art results.
The use of large-scale vision-language models provides a strong prior knowledge for category prediction, enabling the model to generalize well to unseen categories.
The paper provides a thorough evaluation of the model on a new dataset, AVSBench-OV, which is a valuable contribution to the field.

**Limitations:**

The paper assumes that the audio and visual features are aligned in the spatial dimension, which may not always be the case in real-world scenarios.
The model relies heavily on the prior knowledge from large-scale vision-language models, which may not be available or applicable in all scenarios.
The paper does not provide a thorough analysis of the limitations and potential failures of the model, which could be an important aspect of the research.

**Suitability:**

3

---

### Official Review · Reviewer_DPGJ · 2024-05-22

**Rating:** 5
**Confidence:** 3

**Summary:**

The paper introduces a novel framework for open-vocabulary audio-visual semantic segmentation (OV-AVSS), aimed at segmenting and classifying sound-emitting objects in videos across both seen and unseen scenarios. The framework consists of a Universal Sound Source Localization Module (USSLM) for detecting potential sounding objects and an Open-Vocabulary Classification Module (OVCM) that utilizes pre-trained vision-language models like CLIP to classify objects without being confined to pre-defined categories from the training set. Extensive evaluations on the newly created AVSBench-OV dataset demonstrate the model's superior performance, achieving significant improvements over existing methods, especially in recognizing novel categories unseen during training.

**Strengths:**

1. The paper introduces the task and dataset for open-vocabulary audio-visual semantic segmentation, which incorporates audio information to segment objects that have not been previously encountered.

2. The method effectively merges audio-visual fusion with cutting-edge vision-language models by integrating a Universal Sound Source Localization Module with an Open-Vocabulary Classification Module. This combination enables the precise segmentation and classification of a diverse array of sound-emitting objects.

3. The proposed audio-visual baseline model, which generates a class-agnostic mask for each sounding object and correlate the image pixel with the audio segmentation, is applicable to other audio-visual tasks.

**Limitations:**

1. It is misleading to describe the audio-visual fusion module as a solution to the problem of audio entanglement. The audio-visual early fusion module merely extracts categorical features from the sound and aligns them with corresponding image pixels without actually disentangling them. For disentanglement, sound separation methods may be considered.

2. In Table 2, the number of trainable parameters for each model should be explicitly annotated. This will not only highlight the computational overhead introduced by the OV-AVSS strong baseline model but also facilitate a more equitable comparison in subsequent studies.

3. There is a typographical error in the heading of section 2.2, "2.2 Zero-Shot Audio-Viusal Segmentation," where "Viusal" should be corrected to "Visual."

**Suitability:**

3

---

### Official Review · Reviewer_tGjH · 2024-05-23

**Rating:** 6
**Confidence:** 3

**Summary:**

The proposed method introduces an advanced task within the field of audio-visual learning, which extends
traditional audio-visual semantic segmentation (AVSS) to an open-vocabulary setting. This article proposes a new framework,
named OV-AVSS, which integrates a universal sound source localization module with an open-vocabulary classification module
by using the knowledge of pre-trained vision-language models. The approach demonstrates significant improvements on the
AVSBench-OV dataset, achieving higher mIoU scores on both base and novel categories.

**Strengths:**

This paper is well written.

**Limitations:**

In the “Related Work” section, the paper provides detailed comparisons between existing closed-set AVSS methods and the proposed open-vocabulary approach. Additionally, a deeper discussion on the use and advantages of large-scale pre-trained vision-language models in audio-visual tasks could provide better context and understanding of the state-of-the-art methods. A comparison of the performance of FCN-based and Transformer-based methods for both AVSS and broader segmentation tasks would also be beneficial.
The paper could further discuss the limitations of traditional AVSS methods in handling open-vocabulary tasks, including a comparison of their performance against OV-AVSS framework. More experimental evidence could be included to highlight the robustness and scalability of the proposed method across different datasets and scenarios. For instance, detailing the failure modes of the proposed method in different noise conditions or with varying object motion speeds could provide a more comprehensive
analysis.
The paper could also discuss potential challenges and limitations of the universal sound source localization module, such as its performance under varying environmental conditions and with different types of sounds. Furthermore, alternative approaches or enhancements to this module could be explored to provide a more comprehensive solution. The exploration of these limitations can help in understanding the practical implications and areas for future improvement.
In the “Datasets and Metrics” section, it could add specific descriptions and characteristics of the AVSBench-OV dataset to help readers better understand the dataset’s composition and the evaluation metrics used. Detailed statistics on diversity in sound types and the complexity of the scenes could help in better evaluating the model’s performance.
In the “Implementation Details” section, the paper could provide further explanation on how to select model parameters and adjust training strategies to optimize performance. Detailed guidance on hyperparameter tuning, computational resources required, and strategies for efficient training would be beneficial for replicating the results. Providing insights into the training duration, batch size, and learning rate adjustments would also be useful.
In the “Conclusion” section, the paper could propose future research directions and challenges that need to be addressed in the open-vocabulary AVSS field.

**Suitability:**

3

---

### Official Review · Reviewer_tzLT · 2024-05-26

**Rating:** 3
**Confidence:** 3

**Summary:**

This work introduces a new task: open-vocabulary audio-visual semantic segmentation extending audio-visual semantic segmentation(AVSS) task to open-world scenarios beyond the annotated label space. Besides, the paper introduces the first open-vocabulary AVSS (OV-AVSS) framework, which aims to accurately localize and identify any sounding object in a video, assigning semantic labels at the pixel level, regardless of whether the object was include in the training dataset. Quantitative and qualitative experiments on AVSBench-OV dataset show strong zero-shot generalization ability on novel categories unseen during training.

**Strengths:**

The paper is well-organized and written. The concept of open-vocabulary AVSS is innovative as it unveils the shortcomings present in traditional AVSS tasks. The proposed method appears to be rational, and the experimental results have validated the effectiveness of the suggested approach. It harnesses the generalization capabilities of the CLIP model to design the network, endowing the model with robust zero-shot generalization ability, enabling it to segment classes it has never seen before. The research content is innovative and holds potential application value.

**Limitations:**

**As for method.** The reasons behind the capability of the universal sound source localization module to generate class-agnostic masks for unseen classes remain unclear. Regarding universal sound source localization, it utilizes a pre-trained VGGish model to obtain audio representations. VGGish is pre-trained on the AudioSet dataset, showcasing a certain degree of generalization capability. However, this generalization capability does not extend to the image encoder. The question arises as to how training on visible classes enables the alignment of unseen classes with audio representations.

**As for the experimental results.** (1) I believe that the comparative experiments and ablation studies are not sufficiently. Regarding the zero-shot audio-visual segmentation section, the authors only compare their work with GAVS and Sam4AVS. However, GAVS fails to provide specific data, and there is a significant discrepancy between the results of Sam4AVS and OV-AVSS, leading me to question the validity of such comparisons. It remains uncertain whether this comparison adequately demonstrates the superior zero-shot generalization capabilities of the method proposed in the paper. (2) In qualitative analysis, comparisons with other methods can be incorporated, and further analysis of model complexity is recommended.

**Weakness Summary.** The rationality of the model design warrants further analysis. Further investigation should be conducted on why the model possesses the ability to generalize, thereby enabling it to segment unseen classes. In addition, further experiments are expect to be conducted to demonstrate the effectiveness of the model.

**Suitability:**

3

---

### Meta-Review · Area_Chair_wEee · 2024-07-04

**Recommendation:** Accept (Oral)
**Confidence:** 5

**Metareview:**

The paper introduces a new task: open-vocabulary audio-visual semantic segmentation to extend the audio-visual semantic segmentation (AVSS) task to open-world scenarios beyond the annotated label space. It claims that experiments demonstrate the strong segmentation and zero-shot generalization ability of the model in all categories.

The proposed method appears to be rational, and the experimental results have validated the effectiveness of the suggested approach. After considering the paper, the reviewer's comments, and the rebuttal I recommend 'accept' for the paper to enhance its visibility to a wider audience.

The reviewers highlight the following strengths and limitations:

Strengths:
1. The paper is well-organized and written.
2. The concept of open-vocabulary AVSS is innovative
3. The method effectively merges audio-visual fusion with cutting-edge vision-language models by integrating a Universal Sound Source Localization Module with an Open-Vocabulary Classification Module.
4. The use of large-scale vision-language models provides a strong prior knowledge for category prediction, enabling the model to generalize well to unseen categories.

Limitations:
1. Further investigation should be conducted on why the model possesses the ability to generalize, thereby enabling it to segment unseen classes.
2. The misleading description and typographical error (addressed in the rebuttal)